# ADAPTIVE COMPUTATION WITH ELASTIC INPUT SEQUENCE

## ABSTRACT

When solving a problem, human beings have the adaptive ability in terms of the type of information they use, the procedure they take, and the amount of time they spend approaching and solving the problem. However, most standard neural networks have the same function type and fixed computation budget on different samples regardless of their nature and difficulty. Adaptivity is a powerful paradigm as it not only imbues practitioners with flexibility pertaining to the downstream usage of these models but can also serve as a powerful inductive bias for solving certain challenging classes of problems (Dehghani et al., 2018; Banino et al., 2021). In this work, we propose a new strategy, AdaTape, that enables dynamic computation in neural networks via adaptive tape tokens. AdaTape employs an elastic input sequence by equipping an architecture with a dynamic read and write tape. Specifically, we adaptively generate input sequences using tape tokens obtained from a tape bank that can either be trainable or generated from input data. We analyze the challenges and requirements to obtain dynamic sequence content and length, and propose the Adaptive Tape Reading (ATR) algorithm to achieve both objectives. Via extensive experiments on image recognition tasks, we show that AdaTape can achieve better performance while maintaining the computational cost.

## 1 INTRODUCTION

Adaptive computation is central to human intelligence. This is clear, given that humans spend a variable amount of time and energy on different problems depending on their complexity (Meunier et al., 2009). Adaptivity in neural networks is attractive for two key reasons. Firstly, adaptive computation could potentially be a powerful and essential inductive bias for solving challenging problems that would have been significantly harder otherwise (Dehghani et al., 2018; Banino et al., 2021). Secondly, adaptive computation could imbue practitioners with downstream flexibility pertaining to the usage of these models. For the most part, altering the computation budget of a model after it has been trained becomes almost impossible. Hence, the ability to flexibly scale computational costs and budgets dynamically is highly desirable.

This paper proposes AdaTape, a new general-purpose adaptive computation method. The key idea is to introduce elastic input sequences via the means of a dynamic read and write memory tape. Unlike all prior works that investigate adaptivity via sparse conditional computation (Fedus et al., 2022; 2021; Lepikhin et al., 2020) or adaptivity through recursion over architecture (Dehghani et al., 2018; Banino et al., 2021; Graves, 2016), this work presents a new perspective that explores adaptivity with respect to input sequence length (or read/write memory tapes from the perspective of a Neural Turing Machine (Graves et al., 2014)). We postulate that this exploration is crucial for the development of this class of methods and is very complementary to the existing suite of methods developed to encourage adaptive computation in neural networks.

AdaTape promotes adaptivity in both type and amount of computation. Specifically, AdaTape controls (1) the contents of the tape tokens (2) the number of tape tokens, that are used for each input. To this end, AdaTape is characterized by a tape bank that can be dynamically read from, using a newly proposed dynamic halting algorithm which we call Adaptive Tape Reading (ATR). Concretely, ATR method adaptively and dynamically selects the content and length of this memory tape which is appended to the inputs of a standard Transformer (Vaswani et al., 2017). Given that the

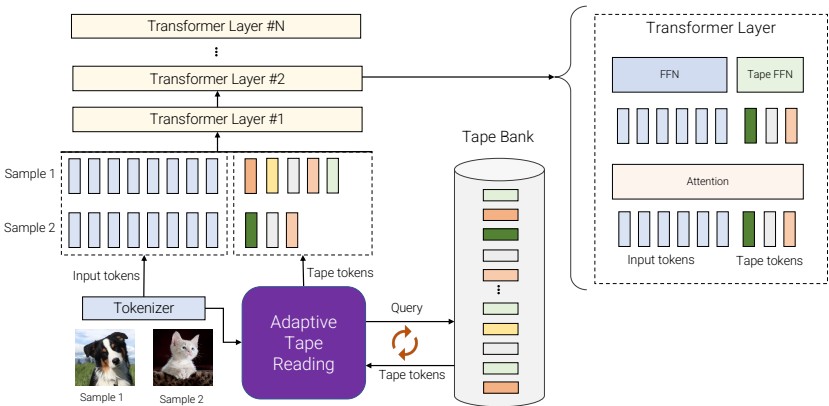

Figure 1: An overview of AdaTape. For different samples, we pick a variable number of different tokens from the tape bank. The tape bank can be driven from input, e.g., by extracting some extra fine-grained information or it can be a set of trainable vectors. The Adaptive Tape Reading is used to recursively select different sequences of tape tokens, with variable lengths, for different inputs. These tokens are then simply appended to inputs and fed to the transformer encoder.

increasing computation budget generally leads to improved quality (Kaplan et al., 2020; Dehghani et al., 2021; Hoffmann et al., 2022; Zhai et al., 2022; Abnar et al., 2021), this enables a new way for adaptively scaling the computation budget without adding new parameters or applying a part of the model recursively.

To ascertain the effectiveness of the proposed AdaTape method, we first evaluate it on the challenging Parity task (Graves, 2016; Banino et al., 2021), a standard verification check for Adaptive Computation Time (ACT) algorithms (Graves, 2016). Our results demonstrate that AdaTape performs well on this problem. Meanwhile, this problem remains completely unsolvable by vanilla Transformers. This not only verifies that the AdaTape inductive bias is crucial in solving certain classes of problems but also asserts its correctness. Given that the standard Transformer is touted as the true universal algorithm with ubiquitous impact across all fields (Jumper et al., 2021; Dosovitskiy et al., 2020; Vaswani et al., 2017), it would be unthinkable if Transformers lack the inductive bias for a standard vector parity problem.

Finally, we conduct large-scale experiments on vision tasks (e.g., image recognition and evaluate their few-shot accuracy), showing that AdaTape performs well and outperforms vanilla Transformers when *compute-matched* (Dehghani et al., 2021) across both FLOPs and throughput. While AdaTape does not improve efficiency during training, the granted flexibility and adaptivity allow dynamic scaling of the computation budget during inference. Given that the standard practice is to train and serve $n$ models to potentially cater to variable computation budgets (e.g., base models for less important workloads and large models for prioritized examples), we consider the property of having a single model flex between multiple requirements to be highly desirable.

## 2  ADATAPE: ADAPTIVE COMPUTATION WITH ELASTIC INPUT SEQUENCE

Neural networks can obtain adaptive ability by using different functions or variable computation budgets for different inputs. Assume a deep neural network is a function $f(x; \theta)$. The output of this function relies on both input $x$ and parameter $\theta$. For adaptive function types, we usually sparsely activate a subset of parameters $\theta$ conditioned on $x$. This type of adaptive ability can also be named as conditional computation. Research on Mixture-of-Experts (Fedus et al., 2021; Lepikhin et al., 2020; Xue et al., 2021; Lou et al., 2021; Riquelme et al., 2021) in fact introduce adaptivity on the function type via routing and specifying the computation for each input sample.

Another line of adaptive computation research is the dynamic computation budget. For standard neural networks (*e.g.,* transformer), the computation budget is fixed for different samples. However, recent studies show that adaptive computation budgets can help to solve many tasks where vanilla transformer totally fails (Dehghani et al., 2018; Banino et al., 2021; Abnar et al., 2020). Most of these works use dynamic depth to achieve adaptivity in the allocation of the computation budget. For instance, the Adaptive Computation Time (ACT) algorithm was proposed by Graves (2016)

---

**Algorithm 1** Adaptive Computation Time

---

1: **Input:** Initial states $\boldsymbol{S} = \{\boldsymbol{s}_1, ..., \boldsymbol{s}_N\}$, where $\boldsymbol{s}_n = \boldsymbol{0}$; Output states $\boldsymbol{S}^{out} = \{\boldsymbol{s}^o 1, ..., \boldsymbol{s}_N^o\}$, where $\boldsymbol{s}_n^o = \boldsymbol{0}$; Initial halting score $h_p = 0$; Initial update times $n = 0$; Max ponder times $T$; Trainable linear layer $g(\cdot)$; Set $\epsilon$ as a small number like 0.01; Initial ponder loss $l_{atr} = 0$.
2: **while** $h_p < 1.0$ and $n < T$ **do**
3:     Compute halting score for each token at this step $\boldsymbol{p} = \text{sigmoid}(g(\boldsymbol{S}))$
4:     Compute the average halting score for each sample $p = \text{mean}(\boldsymbol{p})$
5:     **if** $h_p + p > 1.0 - \epsilon$ **then**
6:         Update remainder $r = 1 - h_p$
7:         Update output states $\boldsymbol{S}^{out}$ += $\boldsymbol{S} * r$
8:         break
9:     Increase update times $n$ += 1
10:     Update states $\boldsymbol{S} = g(\boldsymbol{S})$
11:     Update output states $\boldsymbol{S}^{out}$ += $\boldsymbol{S} * p$
12:     Increase the accumulated halting score $h_p$ += $p$
    $l_{act} = n + r$
13: **return** $l_{act}, \boldsymbol{S}^o$

---

for an adaptive computational budget on recurrent neural networks (Hochreiter & Schmidhuber, 1997). Universal Transformer (UT) (Dehghani et al., 2018) extends the ACT algorithm to transformers (Vaswani et al., 2017) and makes the computation budget rely on the number of transformer layers used for processing each input example or tokens. Many recent studies like PonderNet (Banino et al., 2021) follow the same framework while improving the dynamic halting mechanisms.

AdaTape not only uses different function types per input via conditioning the adaptive tape reading mechanism on the input representation but also adjusts the computation budget by employing variable length memory tape for different inputs. Figure 1 presents a high-level schema of AdaTape in the context of image recognition. For a given batch of input images, after tokenization (e.g., a linear projection of non-overlapping patches from the image in the vision transformer), AdaTape uses a vector representing each input to dynamically select a variable-sized sequence of tape tokens from a tape bank. A tape bank could be a set of trainable vectors or generated on-the-fly from inputs, e.g., by using a more fine-grained tokenizer. The selected tape tokens are appended to the original input and fed to the Transformer layers. More precisely, AdaTape defines $f(\boldsymbol{x}; \boldsymbol{\theta})$, where $\boldsymbol{x} = \boldsymbol{x}^I : \boldsymbol{x}^T$, and $\boldsymbol{x}^I$ being input tokens and $\boldsymbol{x}^T$ being a sequence of tape tokens that is generated by $f_{\text{ATR}}(\boldsymbol{x}^I, \boldsymbol{Z}_{bank})$. When using a learnable bank, $f_{\text{ATR}}$ in fact selects different parameters of the model (learnable tapes) per input, which means AdaTape has adaptivity in terms of the function type. Also given that the length of $\boldsymbol{x}_t$ is different for each input, AdaTape allocates different computation budgets for different inputs. Finally, a sequence containing both input and tape tokens goes to a transformer encoder. For each transformer layer, we use the same multi-head attention across all input and tape tokens. However, we use two different feed-forward networks, one is applied to all tokens from the original input and the other is applied to all tape tokens. We observed slightly better quality by using separate feed-forward networks for input and tape tokens.

In the following section, we briefly introduce the original ACT algorithm and explain its contradiction with the adaptive sequence setting. Then, to satisfy the requirements of the elastic input sequence, we design the Adaptive Tape Reading algorithm, explain the different components in detail, and share the reasoning behind all the design choices.

## 2.1 ADAPTIVE COMPUTATION TIME

As shown in Algorithm 1, ACT uses a trainable linear component with sigmoid activation sigmoid$(g(\cdot))$ that computes the halting score at each step. When the summation of the halting scores $h_p$ is greater than the halting threshold , the computation stops and we conduct early exiting to drop the future layers. At each time step, the states $\boldsymbol{S}$ are weighted by halting score $p$ and accumulated. When the model decides to end the algorithm, the summation of the halting score $p_{sum} = \sum_{t=1}^{T} p_t = 1.0$.

ACT employs a loss function $l_{act}$ to encourage less computation. The loss $l_{act}$ includes two components, the number of updates $n$ and the remainder $r$. The main goal of ACT is to control the computation by minimizing the number of updates $n$. However, such an objective is not differen-

tiable and cannot be minimized directly via back-propagation. To address that in a round-about way, ACT optimizes $n$ together with the remainder $r$. When decreasing $r$, the summation of all used $p_{sum} = \sum_{t=1}^{T} p_t$ increases. When $\sum_{t=1}^{T-1} p_t > 1.0$ - $\epsilon$, $n$ would decrease by 1.0 thus ACT minimizes $n$ indirectly. The final loss function can usually be written as $l = l_{main} + \lambda l_{act}$, where $l_{main}$ represents the main training objective, e.g., cross-entropy loss in the classification task. Note that the stability and performance of the model when using ACT is sensitive to the ACT loss coefficient, so, using ACT requires careful tuning of $\lambda$.

## 2.2 TAPE BANK

AdaTape uses a bank of tokens where all the candidate tape tokens are stored. The adaptive tape reading mechanism is independent of where are the tape tokens from and simply requires interacting with a $\mathbb{R}^{B \times H}$ tensor, where $B$ is the size of the bank and $H$ is the dimension of each token. We explore two different ways of creating the tape bank in this paper, an input-driven bank and a learnable bank.

**Input-driven Bank**  For an input-driven bank, we use input data as the source to generate bank tokens on-the-fly. The general idea is to extract a bank of tokens from the input, while employing a different approach than what the model tokenizer uses for mapping the raw input to a sequence of input tokens. This enables dynamic, on-demand access to information from the input that is obtained using a different point of view, e.g., a different resolution or a different level of abstraction. For instance, for the image recognition task, when using Vision Transformers (Dosovitskiy et al., 2020), a simple, yet effective idea is to use different patch sizes for generating input tokens and bank tokens. For instance, using a smaller patch size for generating the bank can be seen as dynamic multi-scale processing of the input where we can select what fine-grained information from the input is useful which is much more efficient than using all the small patches and consuming a large amount of computation. Appendix A.3 provides more details on input-driven bank setup for Vision Transformer. In this paper, we use the input-driven bank in one of our image recognition experiments, however, the idea can be generalized to other tasks and setups, e.g., language tasks where more fine-grained tokenization (Kudo & Richardson, 2018) has shown to be effective.

**Learnable bank**  Using a finer-grain tokenizer for generating a tape bank is not always applicable. For instance, it is hard to further split each node in graph transformer (Yun et al., 2019). A different and more general approach for generating the tape bank, is to simply use a set of trainable vectors as tape tokens. The learnable bank can be seen as an embedding layer with a trainable tensor of size $\boldsymbol{Z}_{bank} \in \mathbb{R}^{B \times H}$. One difference between an input-driven bank is, instead of building the bank on-the-fly, the content is fixed for different samples. On the other hand, selecting tape tokens from a learnable bank means partially using the parameters of the model, which is a form of sparsity in AdaTape.

## 2.3 ADAPTIVE COMPUTATION TIME FOR ELASTIC INPUT SEQUENCE

In order to have elasticity on the input length, we need a mechanism that selects a variable sequence of tape tokens from the bank conditioning on the input representation. Such a mechanism needs to be able to make a decision on how many tape tokens should be appended to the input, e.g., using a halting mechanism where tape tokens are recursively added to the input. As the input representation, we can use `[CLS]` token or the average pooling of all input tokens. This input query vector is used for making the decision on the number of tape tokens for each input.

One straightforward method to obtain adaptive sequence is adapting ACT algorithm explained in Section 2.1. To this end, we first summarize the requirements of ACT algorithm as follows: (1) the summation of the halting score $p_{sum} = \sum_{t=1}^{T} p_t = 1.0$; (2) each halting score $p_t$ is proportional to the importance of the current step output; (3) the computing process of halting score $p_t$ is required to be differentiable and can be well-trained.

However, unfortunately, all these requirements in ACT are not desirable in the adaptive sequence scenario. First, in adaptive token reading, the output state of $t^{\text{th}}$ step is $t^{\text{th}}$ tape token we are going to use as the model input. For the steps that we do not want to stop, we expect the weights of the tokens to stay close to 1.0. On the other hand, for the step we want to stop at, which means this incoming

---

**Algorithm 2** Adaptive Tape Reading

---

1: **Input:** Initial halting score $h_p = 0$; A list of tokens selected $states = []$; Initial query $\boldsymbol{q} \in \mathbb{R}^{1 \times h}$; Tape token bank $\boldsymbol{Z}_{bank} \in \mathbb{R}^{C \times H}$; Token bank mask $\boldsymbol{m} = \boldsymbol{0}$, where $\boldsymbol{m} \in \mathbb{R}^{1 \times C}$, where $\boldsymbol{m} = \boldsymbol{0}$; Max ponder times $T$; Halting threshold $\tau$; Initial ponder loss $l_{atr} = 0$.
2: **while** $h_p < \tau$ **do**
3:     Compute the inner product of query and tape tokens $\boldsymbol{d} = \boldsymbol{q} \cdot \boldsymbol{Z}_{bank}[:, :h]^T$, where $\boldsymbol{d} \in \mathbb{R}^{1 \times C}$
4:     Mask inner product $\boldsymbol{d} = \boldsymbol{d} \odot (1.0 - \boldsymbol{m})$
5:     Compute index of top $K$ elements in $\boldsymbol{d}$, set the index vector as $\boldsymbol{i} \in \mathbb{R}^{1 \times K}$, where $K = \frac{T}{\tau}$
6:     Take top $K$ elements from $\boldsymbol{d}$ by the index vector $\boldsymbol{i}$ and denote this new vector as $\boldsymbol{w}$, where $\boldsymbol{w} \in \mathbb{R}^{1 \times K}$
7:     Normalize the elements by softmax: $\boldsymbol{w} = \text{softmax}(\frac{\boldsymbol{w}}{\sqrt{h}})$
8:     Take top $K$ tape tokens from $\boldsymbol{Z}_{bank}$ by the index vector $\boldsymbol{i}$, and merge the tape tokens $\{\boldsymbol{s}_1, ..., \boldsymbol{s}_K\}$ selected from bank as one single tape token $\boldsymbol{s} = \sum_{k=1}^{K} w_k \boldsymbol{s}_k$
9:     Append this token into input sequence $states$ += $\boldsymbol{s}$
10:    **if** $h_p + \max(\boldsymbol{w}) > \tau$ **then**
11:       break
12:    Increase halting score: $h_p$ += $\max(\boldsymbol{w})$
13:    Increase ponder loss: $l_{atr}$ += $1.0 - \sum_{k=1}^{K} w_k^2$ or $l_{collect}$ += $h_p$
14:    Update token mask by the one-hot vector of $\boldsymbol{i}$: $\boldsymbol{m}$ += $\text{sum}((\text{one-hot}(\boldsymbol{i}), \text{axis}=0)$
15:    Update query $\boldsymbol{q} = \frac{1}{2}(\boldsymbol{s} + \boldsymbol{q})$ or $\boldsymbol{q} = \boldsymbol{s}$
16: **return** $l_{atr}, states$

---

tape token is not as important as others, we need a smaller weight than previous tokens. Such a requirement totally contradicts to requirements (1) and (2) in ACT. Additionally, the existence of layer normalization $\text{LayerNorm}(\cdot)$ will make trainable halting score computation layer $g(\cdot)$ in ACT algorithm invalid. The main reason is the normalization layer will ignore the halting score $p_t$: $\text{LayerNorm}(p_t \boldsymbol{z}_t) \approx \text{LayerNorm}(\boldsymbol{z}_t)$. The halting score computation layer $g(\cdot)$ will then cannot be well-trained. We introduce the detailed reasoning in Appendix A.4. In summary, after analyzing the requirements of ACT algorithm and adaptive sequence length, we found there are some clear contradictions. Therefore, we devote ourselves to designing a novel dynamic halting algorithm in the following section.

## 2.4 Adaptive Tape Reading Mechanism

Algorithm 2 presents a pseudo code for the Adaptive Tape Reading. ATR is an iterative process for selecting tape tokens from the bank. At each iteration, we select a new set of tokens (here we select and add tape tokens one-by-one) without replacement, conditioned on the previously selected tokens. ATR uses a query vector $\boldsymbol{q} \in \mathbb{R}^H$ representing the input at the current iteration (i.e., the sequence of all input tokens plus already selected tape tokens) to select the next set of tokens from a tape bank $\boldsymbol{Z}_{bank} \in \mathbb{R}^{B \times H}$. We compute $\boldsymbol{d}$ as the inner product of $\boldsymbol{q}$ and $\boldsymbol{Z}_{bank}$. Note that in practice, to reduce the computation, we can use part of the $\boldsymbol{q}$ and tape tokens for computing the inner product (e.g., $\boldsymbol{q}[: h]$, and $\boldsymbol{Z}_{bank}[:, :h]$, meaning to use the first $h$ elements). This can be seen as using first $h$ elements in $\boldsymbol{q}$ and $\boldsymbol{Z}_{bank}[b, :h]$ as key and the remaining elements as value, where $\boldsymbol{Z}_{bank}[b, :]$ denotes $b^{\text{th}}$ tape token. To avoid the repeated selection of tape tokens, at each iteration, we adjust the inner product $\boldsymbol{d}$ by masking out weights of tokens that are selected before (using the bank mask $\boldsymbol{m}$ in Algorithm 2 that gets updated in each iteration). Then we select top-$K$ tape tokens according to their scores from $\boldsymbol{d}$ and create $\boldsymbol{w} \in \mathbb{R}^{1 \times K}$ containing weight logits of the selected tape tokens in form. We normalize $\boldsymbol{w}$ and apply softmax. We also create the corresponding tape tokens vectors $\{\boldsymbol{s}_1, ..., \boldsymbol{s}_K\}$ collected from the $\boldsymbol{Z}_{bank}$, and compute a single vector $\boldsymbol{s}$ as the weighted average of the selected $K$ tokens, which is the tape token that we add to the input at the current iteration.

To make the halting decision, we accumulate the largest value in $\boldsymbol{w}$ into $h_p$ until it is greater or equal to a threshold $\tau$. Larger $\max(\boldsymbol{w})$ indicates the need for less iteration (i.e., adding fewer tape tokens). Note that the query vector $\boldsymbol{q}$ used for selecting the tape token is updated by averaging the input query and current tape token or just replacing the input query with the current tape token.

In order to incentivize shorter sequences for efficiency and penalize the model for adding tape tokens when there is no need, we use a similar loss term to what the original ACT uses, i.e., $l = l_{main} + \lambda l_{atr}$. We observe, unlike ACT, the adaptive tape reading is less sensitive to the value of $\lambda$. We

also empirically found that, when using a learnable bank, we can omit the $l_{atr}$ loss term without observing a significant change in the performance.

ATR offers a template for elastic input sequences. To emphasize some of the details in the above algorithms: (1) we observe better performance and stability when using a lower dimension for the query; we found normalizing the inner product by query dimension $\sqrt{h}$ before applying softmax critical following the original design of attention (Vaswani et al., 2017); (3) we found that normalizing the bank and query using a shared LayerNorm$(\cdot)$ layer improves performance and stabilizes training.

## 3  EXPERIMENTS

We first present our experimental setup and implementation details in Section 3.1. We report the results on the parity task and image classification benchmarks in Section 3.2 and 3.3. To further validate the effectiveness of our model, we conduct an ablation study in Section 3.4. Finally, Section 3.5 provides some insights into the behavior of adaptive tape reading in AdaTape.

### 3.1  EXPERIMENTAL SETTING

We evaluate AdaTape on both parity task and image recognition benchmarks. First, the parity task is, given a sequence of numbers $x = [x_1, x_2, ..., x_N]$, where $x_n$ is 1, $-1$, or 0, we are to predict whether the number of 1's in $x$ is even or odd. For the parity task, we employ an input-driven bank and use the bank as the way the model accesses the input information, and as the input token, $s$, we simply use a single trainable token vector (*e.g.,* `[CLS]` token) to create the initial query for tape token reading. We train all models for 10000 steps with batch size 128. The learning rate is set as 3e-5, and we use a linear warm-up for 1000 steps. We use transformer-Tiny in this task since we cannot see improvements when we scale it up, similar to the observation of UT (Dehghani et al., 2018) for algorithmic tasks. The configuration of transformer-Tiny can be found in Appendix A.5.

For image classification benchmarks, we first conduct large-scale pre-training on JFT-300M (Sun et al., 2017) followed by few-shot learning on a wide range of datasets, including ImageNet (Deng et al., 2009), Cifar100 (Krizhevsky et al., 2009) and Pets (Parkhi et al., 2012) following the protocol of vanilla ViT (Dosovitskiy et al., 2020) and Big Transfer (Kolesnikov et al., 2020). That is, we use the hidden representation before logits computation as a feature to adapt the upstream model, and evaluate the resulting model on the validation/test set. We also train models on ImageNet from scratch for easier comparison in future work. Following existing work on ViT with adaptive computation (Yin et al., 2022), on ImageNet, we train models mainly at Tiny and Small scales. Please see Appendix A.6 and Appendix A.7 for details on hyper-parameters and training strategies used for AdaTape with a learnable bank.

We compare AdaTape with standard transformers and adaptive transformers. The standard transformers include ViT (Dosovitskiy et al., 2020), DeiT (Touvron et al., 2021) and PlainViT (Beyer et al., 2022). DeiT and PlainViT are heavily-optimized models for training on ImageNet from scratch. We also compare with adaptive transformers like UT (Dehghani et al., 2018) and A-ViT (Yin et al., 2022). To further enhance our baselines and achieve a more comprehensive comparison, we develop two improved versions of UT as our strong baselines equipped with adaptive computation. We first consider UT without parameter sharing and refer to it as Unshared Universal Transformer (U2T). We increase the maximum number of pondering steps in U2T, which means U2T has more layers, more computation and more parameters. To further enhance the baselines, we also consider stacking UT and ViT together. We found putting a shallow UT on top of ViT totally failed due to an unstable training process, but putting ViT on top of UT can achieve decent performance. We refer to this model as UViT which is our strongest baseline on image classification with adaptive depth. In UViT, we employ 3 UT layers on Tiny, Small, and Base models and 6 UT layers on Large models. The number of standard transformer layers is the same as ViT.

### 3.2  EVALUATION ON THE PARITY TASK

We evaluate AdaTape on the parity task to study the effect of the inductive biases that it introduces to vanilla Transformer on a synthetic task. The Parity is the simplest non-counter-free, or periodic

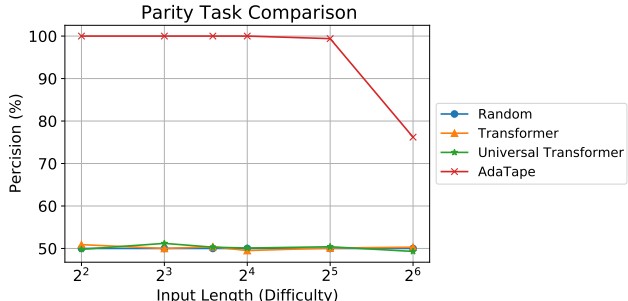

Figure 2: Evaluation on the parity task. The standard transformer and universal transformer totally failed on this task, both showing performance at the level of a random guessing baseline.

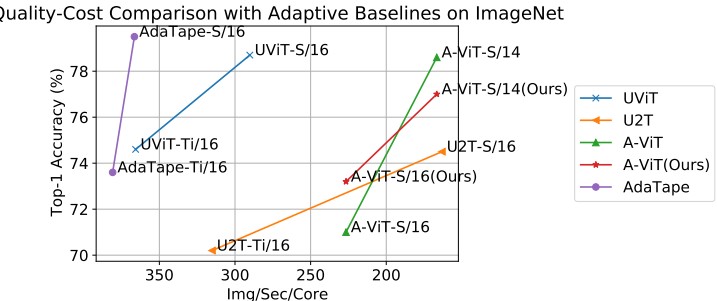

Figure 3: We evaluate AdaTape by training on ImageNet from scratch. For A-ViT, we not only report their results from the paper but also re-implement A-ViT by training from scratch, *i.e.,* A-ViT(Ours).

regular language (Bhattamishra et al., 2020). Simple recurrent neural networks can solve this task well because the memory in the recurrent neural network can record the states for finite-state automation (Abnar et al., 2021; Schwarzschild et al., 2021; Veličković et al., 2022; Ibarz et al., 2022; Bansal et al., 2022). Standard transformer totally failed in modeling such sequences (Hahn, 2020) as they are incapable of directly maintaining a counter. Specifically, in AdaTape, we can use the tape tokens to record the behavior in each recurrence step. We first use the input sequence $x$ to generate the bank. The Adaptive Tape Reading algorithm plays the role to model the parity task recurrently. Considering the target is to predict the number of $1$ in the input sequence is even or odd, there are actually two states in the finite-state automation. We then fix the $K$ as 2 in Algorithm 2 to check whether there is a state switching after each step. When $K$ is fixed as 2, the maximum ponder time $T=2\tau=\frac{N}{2}$, where $N$ is the length of input parity sequence. Such a hyper-parameter setting is to match the period in parity and to ensure AdaTape can collect all required tokens to make the prediction.

Figure 2 shows the results on the parity task. We can see both the standard transformer and universal transformer completely fail on the parity task, even if they are trained and evaluated on a very short (simple) sequence. However, the AdaTape performs much better than all baselines. One reason is that AdaTape incorporates the recurrence within the model input selection and such an inductive bias could be of crucial for solving the parity task, as it is a way to implicitly enables AdaTape to maintain a counter, which is not possible in the standard transformer.

### 3.3 EVALUATION ON IMAGE CLASSIFICATION

We pre-train AdaTape on JFT-300M and report few-shot results on popular image recognition benchmarks in Table 1. We can see U2T performs worse than other models, we suggest the reason is the unstable training loss when making new halting decisions. The advanced baseline, UViT performs better than ViT because of adaptivity, more computation, and parameters. Our AdaTape is using less computation than UViT and performs better at all scales. AdaTape with an input-driven bank is superior at a larger scale. We suggest a larger scale model is better at finding informative tokens from input and validate this reasoning in Section 3.5. We can also observe AdaTape is not very good at throughput. Both types of AdaTape are not as hardware friendly as baselines. This is the limitation of AdaTape. However, please note this only means the training speed is slightly slower on TPU, which is highly optimized by large-scale matrix multiplication. Due to less real computation (GFLOPs) and dynamic sequence length, AdaTape has the potential to speed up inference on other hardware or smaller batch size.

Table 1: Pre-training and transfer learning results on image classification benchmarks. We report two AdaTape models with different bank types at each scale. AdaTape-B/32-Learn means we are using a learnable bank for AdaTape-B/32. AdaTape-B/32-Input denotes we are using an input-driven bank. We report both GFLOPs and throughput (measured in images / second / core). For all models with adaptive computation budget, we report the upper bound computation cost. The pre-training is conducted on JFT-300M dataset and we report precision@1 (%) on the validation dataset. The few-shot experiments are on ImageNet, Cifar100, and Pets datasets with Top-1 accuracy. $IN_{25}$ denotes the result on ImageNet 25-shot.

| Model | GFLOPs | Throughput | JFT-300M | $IN_{10}$ | $IN_{25}$ | $Cifar100_{10}$ | $Pets_{10}$ |
|---|---|---|---|---|---|---|---|
| ViT-B/32 | 4.437 | 793.8 | 43.3 | 59.4 | 62.5 | 72.2 | 90.3 |
| U2T-B/32 | 5.513 | 343.4 | 38.8 | 49.5 | 53.6 | 64.4 | 84.1 |
| UViT-B/32 | 5.511 | 478.7 | 44.4 | 61.9 | 64.6 | 74.4 | 91.6 |
| AdaTape-B/32-Learn | 5.585 | 431.8 | 44.4 | 63.0 | 65.8 | 75.0 | **93.2** |
| AdaTape-B/32-Input | 6.057 | 342.3 | **44.7** | **63.8** | **65.9** | **75.5** | 93.0 |
| ViT-B/16 | 17.634 | 253.7 | 48.6 | 67.4 | 70.0 | 77.5 | 93.0 |
| U2T-B/16 | 22.016 | 102.9 | 46.6 | 63.0 | 66.0 | 73.1 | 92.5 |
| UViT-B/16 | 22.011 | 151.7 | 49.0 | 68.5 | 71.0 | 75.8 | 94.0 |
| AdaTape-B/16-Learn | 18.837 | 167.1 | **49.1** | **70.3** | **72.6** | **78.7** | 94.0 |
| AdaTape-B/16-Input | 19.304 | 148.8 | 48.9 | 69.0 | 71.5 | 77.4 | **94.4** |
| ViT-L/32 | 15.628 | 192.3 | 49.9 | 69.5 | 72.1 | 79.4 | 94.0 |
| UViT-L/32 | 19.242 | 127.1 | **50.5** | 70.5 | 72.6 | 80.1 | 94.1 |
| AdaTape-L/32-Learn | 19.497 | 108.6 | 50.2 | 70.8 | 73.3 | 79.6 | **95.0** |
| AdaTape-L/32-Input | 20.141 | 95.5 | 50.2 | **71.8** | **73.8** | **81.3** | **95.0** |
| ViT-L/16 | 61.724 | 63.1 | 54.3 | 75.2 | 77.0 | 83.1 | 95.9 |
| UViT-L/16 | 77.114 | 44.7 | **54.8** | 75.5 | 77.4 | 81.6 | 95.7 |
| AdaTape-L/16-Learn | 65.941 | 44.2 | 54.7 | 76.5 | 78.0 | 82.7 | **96.7** |
| AdaTape-L/16-Input | 66.570 | 40.1 | **54.8** | **76.7** | **78.5** | **84.7** | 96.4 |

We also evaluate AdaTape by training on ImageNet-1K from scratch. For A-ViT (Yin et al., 2022), authors fine-tuned their model to obtain adaptivity from a pre-trained ViT checkpoint, which has a different setting with ours. For a straightforward comparison, we not only report their results from the paper but also re-implement A-ViT to reproduce the results when training from scratch. The quality-cost curve is shown in Figure.3. We can see AdaTape performs much better than the baselines in terms of quality-cost balance. For efficiency, AdaTape-S/16 is even faster than the Tiny-level baselines. Such results match well with the finding from Dehghani et al. (2021), which shows that the adaptive model depth architectures are not well suited for many accelerators like TPU. AdaTape is much more efficient compared to other adaptive baselines because we are injecting adaptivity into the input sequence instead of model depth. For effectiveness, AdaTape can even outperform the non-adaptive models, ViT and DeiT, with comparable computation costs. More detailed results can be found in Appendix A.8

## 3.4 ABLATION STUDY

We first ablate the adaptive abilities of AdaTape including adaptive sequence content and adaptive sequence length. For adaptive sequence content, we expect it can improve effectiveness. For the adaptive sequence length, as it has the potential to save computation during inference, we expect AdaTape can keep a comparable accuracy when we use the fixed sequence length.

Adaptive sequence length is from ATR algorithm with a recurrent token selection process. The adaptive sequence content is mainly from a selective use of the tape bank. To ablate the model by removing only the "adaptivity in length", we set the halting threshold to infinite high and always select $T$ tokens for different samples, which means all samples use the maximum sequence length and the same amount of computation costs. Note that models without adaptive sequence length use the upper bound of computation used by the adaptive ones. Different from adaptive sequence length, removing only the "adaptivity of content" is not possible, because our ATR algorithm at the end of the day uses a tape bank (which is inherently adaptive in content). Therefore, we remove the "adaptivity in length and content" together by appending a fixed set of trainable tokens to every input sample.

Table 2: Ablation study on the adaptive abilities of AdaTape. We use AdaTape with an input-driven bank as a platform. For the model without adaptive length, we pick $T$ tape tokens in parallel. For the model without adaptive content and adaptive length, we remove the tape bank and use a fixed set of trainable tokens to enhance the input. We also report average/max sequence length and sequence length variance.

| Model | JFT-300M | IN 10-shot | IN 25-shot | Avg/Max SeqLen | Var SeqLen |
|---|---|---|---|---|---|
| AdaTape-B/32 | 44.7 | 63.8 | 65.9 | 57.4/59.0 | 4.8 |
| w/o Ada Length | 44.2 | 63.3 | 66.4 | 59.0/59.0 | 0.0 |
| w/o Ada Length&Content | 43.9 | 61.0 | 64.1 | 59.0/59.0 | 0.0 |
| AdaTape-B/16 | 48.9 | 69.0 | 71.5 | 203.4/206.0 | 7.3 |
| w/o Ada Length | 49.2 | 69.4 | 71.8 | 206.0/206.0 | 0.0 |
| AdaTape-L/32 | 50.2 | 71.8 | 73.8 | 56.8/59.0 | 8.5 |
| w/o Ada Length | 50.5 | 71.7 | 74.2 | 59.0/59.0 | 0.0 |
| AdaTape-L/16 | 54.8 | 76.7 | 78.5 | 203.0/206.0 | 10.4 |
| w/o Ada Length | 54.5 | 76.7 | 78.4 | 206.0/206.0 | 0.0 |

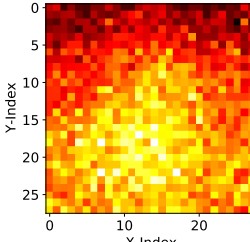 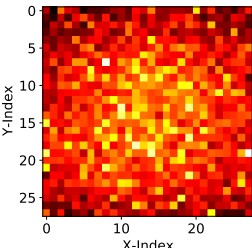

Figure 4: We visualize the tape token selection heatmap of AdaTape-B/32 (left) and AdaTape-B/16 (right). The hotter color means the patch at this position is more frequently selected.

Results are shown in Table 2. We can see, without the adaptive content, there is a significant performance drop. For instance, compared with AdaTape without adaptive sequence only, AdaTape-B/32 without adaptive content degrades 2.3 points on ImageNet 10-shot accuracy. This shows the effectiveness of adaptive sequence content. When we remove the adaptive sequence length, we can see models perform comparably instead of much better at all scales, which shows the tape tokens selected are condensed and make full use of limited input tokens. Another interesting finding is, when we scale the model up, we can observe the increasing sequence length variance. This indicates larger AdaTape is better at controlling sequence length and computation cost. We also conduct the ablation study on the hyper-parameters, computation cost, and the number of trainable parameters in Appendix A.6, A.10 and A.11, respectively.

### 3.5 VISUALIZATION

In this section, we study the behavior of AdaTape by visualization. First, we collect the token selection results of AdaTape with an input-driven bank on JFT-300M validation set, and visualize them as heatmaps in Figure 4. We can see the central patches are more frequently picked (with lighter colors). This matches well with our prior knowledge, central patches are usually more informative. This shows AdaTape prefers more informative patches to improve performance. We also visualize the token selection distribution. Please see Appendix A.12 for details.

## 4 CONCLUSION

We propose AdaTape, a novel strategy and a general-purpose adaptive computation method. AdaTape is characterized by elastic sequence lengths generated from a newly proposed Adaptive Tape Reading mechanism. This new inductive bias enables AdaTape to have the potential to solve challenging tasks that standard transformers and existing adaptive architecture transformer struggle at. Via extensive experiments on image recognition benchmarks, we show that AdaTape outperforms standard transformers and adaptive architecture transformers in a compute-matched setup.

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

# A APPENDIX

## A.1 RELATED WORK

### A.1.1 DYNAMIC HALTING ALGORITHM

Adaptive Computation Time (ACT) algorithm (Graves, 2016) was first proposed for dynamic halting in recurrent neural networks. Dehghani et al. (2018) adapted this algorithm to transformer-based models and proposed universal transformer. Universal transformer with shared trainable parameters across transformer blocks has adaptive depth for different samples, and this work shows that the adaptivity enables the transformer to solve some tasks that the vanilla transformer totally failed. Yin et al. (2022) adapted ACT algorithm to the vision transformer along the depth. However, the ACT-based transformer training is not as stable as the vanilla transformer and it needs careful hyper-parameters tuning. To overcome this weakness, PonderNet (Banino et al., 2021) improves the universal transformer by reformulating the halting policy as a probabilistic model. Balagansky & Gavrilov (2022) further stabilizes this training process by removing the sampling process in Ponder-Net. Another way to halt adaptively is confidence-based dynamic halting algorithms. Schuster et al. (2021; 2022); Schwartz et al. (2020) investigated the confidence-based dynamic halting algorithms on adaptive model depth.

Our work also proposed a dynamic halting algorithm, but we are interested in pondering at input sequence length instead of model depth. Another difference is that we have no trainable layers to compute the halting score, which is a requirement of adaptive transformer input. Alternatively, we use the entropy of logits from the token selection layer as an indicator to make a halting decision.

### A.1.2 ADAPTIVE SEQUENCE LENGTH

Adaptive sequence length can be achieved by token pruning. For instance, Meng et al. (2022) prune the patch tokens in the vision transformer via a light-weight decision network, and Fayyaz et al. (2022) drop the patch tokens by sampling adaptively. Our AdaTape is different from their work as AdaTape appends a small number of tokens to the original input adaptively instead of pruning the intermediate token representations within the model. Also, the content of the tokens in the existing work is not adaptive. The appended tokens in AdaTape are sparsely selected from the tape bank, which is helpful to improve the effectiveness of the transformer.

Wang et al. (2021) proposed another way to achieve adaptive sequence length for ViT. The proposed model uses a large patch size at first and decreases the patch size adaptively for different samples. This is similar to our input-driven bank. However, first, in our input-driven bank, instead of using all smaller patches, we use only a subset of the fine-grained patches adaptively. In addition, instead of limiting the setup to image-only inputs, AdaTape supports a learnable bank and formulates this problem in a more general way.

### A.1.3 EXTRA TOKENS

Extra tokens are another important component of AdaTape. Song et al. (2021) use an extra special token for vision transformer-based object detection. Burtsev et al. (2020) introduces memory tokens to augment transformer capacity. Using learnable prompt tokens in NLP (Lester et al., 2021) can be also perceived as adding extra tokens to the input. In all these works, the number of added tokens is fixed across different samples and often uses a deterministic set of tokens (imagine having more than one `[CLS]` token). However, AdaTape selects and appends a variable number of tokens from the bank adaptively per sample. Such an adaptive input sequence provides not only a larger capacity but also a flexible computation budget.

## A.2 INPUT-DRIVEN BANK DETAILS

We introduce more details about the input-driven bank of AdaTape. We generate the input-driven bank by:

$$\boldsymbol{Z}_{bank} = h_2(h_1(\boldsymbol{X}_p) + \boldsymbol{E}_{pos}) \tag{1}$$

where $\boldsymbol{X}_p$ is the input sequence with fine-grained tokenization (*e.g.*, smaller patch size for ViT), $h_1$ and $h_2$ are both trainable linear projection, $\boldsymbol{E}_{pos}$ is position embedding. Since $\boldsymbol{Z}_{bank}$ is conditioned on the input sample, we need to generate the content of the bank on-the-fly. This is also the reason why an input-driven bank is slightly more computationally expensive than a learnable bank.

### A.3  SCALING WITH PATCH SIZE

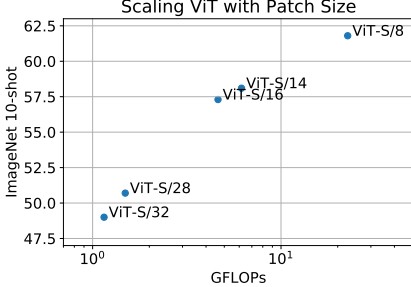

Figure 5: We scale ViT with patch size. The model is pre-trained on JFT-300M (Sun et al., 2017). We report the few-shot performance on ImageNet (Deng et al., 2009)

We fix all other parameters and scale ViT with a smaller patch size to check the effectiveness of smaller patches. We experiment with ViT-S which will not become expensive as the sequence length grows. For example, training ViT-B/8 with data parallelism on 512 TPUv3 cores has the OOM issue. In Figure 5, when adding more computation, although there is a significant accuracy improvement, but more patches introduce very expensive computation costs. Such a trade-off inspires us to use smaller patches as the source to generate bank for a efficient fine-grained data modeling.

### A.4  LAYERNORM IS MAKING ACT INVAILD IN ADAPTIVE SEQUENCE

We argue that naively applying layer normalization in the transformer invalidates the ACT algorithm in AdaTape. To justify this, we first assume we are using a transformer with the following standard transformer architecture:

$$\boldsymbol{X}' = \text{MSA}(\text{LayerNorm}(\boldsymbol{X})) + \boldsymbol{X} \tag{2}$$

$$\boldsymbol{X}'' = \text{FFN}(\text{LayerNorm}(\boldsymbol{X}')) + \boldsymbol{X}' \tag{3}$$

where $\text{MSA}(\cdot)$ is multi-head attention layer and $\text{FFN}(\cdot)$ is feed-forward network. We are going to replace $\boldsymbol{X}$ by $p_t \boldsymbol{z}_t$, where $p_t$ is the halting score of $t^{\text{th}}$ tape token $\boldsymbol{z}_t$. Then, the output of first $\text{LayerNorm}(\cdot)$ should be:

$$\text{LayerNorm}(p_t \boldsymbol{z}_t) = \frac{p_t \boldsymbol{z}_t - \frac{1}{H} \sum_{h=1}^{H} p_t z_{t,h}}{\sqrt{\frac{1}{H} \sum_{h=1}^{H} (p_t z_{t,h} - \sum_{j=1}^{H} p_t z_{t,j})^2 + \epsilon}} * \gamma + \beta \tag{4}$$

since $\epsilon$ is a very small constant, it is reasonable to write the equation above as:

$$\text{LayerNorm}(p_t \boldsymbol{z}_t) \approx \frac{\boldsymbol{z}_t - \frac{1}{H} \sum_{h=1}^{H} z_{t,h}}{\sqrt{\frac{1}{H} \sum_{h=1}^{H} (z_{t,h} - \sum_{j=1}^{H} z_{t,j})^2 + \epsilon}} * \gamma + \beta = \text{LayerNorm}(\boldsymbol{z}_t) \tag{5}$$

We can see $p_t$ is ignored during the normalization, which means the value of $p_t$ cannot change the output of the first $\text{LayerNorm}(\cdot)$ in practice. Since $p_t$ is the output of the trainable layer $g(\cdot)$ in ACT, the $g(\cdot)$ cannot be really trained well by back-propagation. To alleviate this issue, one possible solution is applying the weight $p_t$ to tape tokens after normalization layers. This is a simple and straightforward solution that may make sense. However, empirically, we observe the weights of all tokens will increase to 1.0 very fast in practice and the model will then always only append one token even if we do not use any loss function to penalize longer sequence. We suggest the reason is

that uniformed token mean and variance are highly desired by attention layers. Anyhow, this result is not expected because there is no adaptive ability observed. In addition, we observed the training is also extremely unstable under this design.

In summary, we cannot apply scalar weight to the tape token, so we cannot have a trainable linear layer to compute the halting score $p$. That makes the ACT-based dynamic halting mechanism, including PonderNet, not applicable to AdaTape. We, therefore, are required to design a new adaptive computation algorithm for elastic input sequence, *i.e.,* Adaptive Tape Reading. Such a reasoning process can also be extended to other future conditional and adaptive computation work. For instance, if you want to feed the layer norm with one token after applying weight on the token, you need to be careful about where is the weight from. If this weight is from a trainable layer like ACT, we must check whether this layer can be well-trained via reasoning.

### A.5    CONFIGURATION FOR TRANSFORMER-TINY

Table 3: Transformer configuration for all tiny-level models.

|                  | Transformer-Tiny |
| ---------------- | ---------------- |
| Depth            | 12               |
| Hidden Dimension | 196              |
| MLP Dimension    | 768              |
| #Attention Heads | 3                |

We use the tiny configuration for all models on the parity task. As shown in Table 3, the Tiny level transformer still uses 12 layers, which is the same as the transformer base. The difference is that tiny configuration has smaller hidden dimensions and fewer attention heads.

### A.6    HYPER-PARAMETERS

Table 4: Hyper-parameters for AdaTape on image classification

|                         | AdaTape-Learn | AdaTape-Input      |
| ----------------------- | ------------- | ------------------ |
| Max ponder times $T$    | 10            | 10                 |
| Halting threshold $\tau$ | 2.0          | 2.0 (B) /1.0 (L)   |
| Loss weight $\lambda$   | 0             | 0.01               |
| Bank Size $C$           | 10000         | 784                |

On JFT-300M pre-training, we follow Dosovitskiy et al. (2020) and train all models for 7 epochs. We use the same learning rate, batch size, and learning schedule. Customized hyper-parameters for AdaTape are summarized in Table 4. We employed the fixed max ponder times for all models. Smaller $\tau$ on AdaTape-L with an input-driven bank. The bank size is 10000 for AdaTape-learn. We use bank size 784 for AdaTape Input as we set patch size as 8 to generate tokens from images with 224×224 resolution. AdaTape with a learnable bank can be trained without halting loss. Also, note that we append tape tokens after first transformer encoder layer for better query quality and tape selection.

For ImageNet training from scratch, we summarized the data augmentation and corresponding hyper-parameters in Table 5. Similar with existing work (Beyer et al., 2022), we used Mixup (Zhang et al., 2017), RandAug (Cubuk et al., 2020) and label smoothing (Szegedy et al., 2016) to improve the robustness.

### A.7    TRICKS OF LEARNABLE BANK TRAINING

We found the training of AdaTape with a learnable bank is relatively unstable. To alleviate this, we propose two tricks to improve the training process. The core idea of these two tricks is to improve the diversity of tape tokens and encourage the model to explore the tape bank. We first add noise

Table 5: Hyper-parameters when training on ImageNet-1K only. Ti, S and B denote Tiny, Small, and Base scales.

| Name | Value |
|---|---|
| Learning Rate | 0.001 |
| Linear Warmup Steps | 10000 |
| Learning Rate Decay | Cosine Decay |
| Optimizer | AdamW |
| $(\beta_1, \beta_2)$ | $(0.9, 0.999)$ |
| Weight Decay | 1e-4(Ti), 8e-5(S&B) |
| Epoch | 300 |
| Batch Size | 1024 |
| Mixup | 0.2(Ti), 0.5(S&B) |
| Label Smoothing | 0(Ti), 0.1(S&B) |
| RandAug | (2,10)(Ti), (2,15)(S&B) |

to query $q = q + \lambda \epsilon$, where $\epsilon$ is sampled from standard normal distribution and $\lambda$ is the weight of noise. We set $\lambda$ as 0.01 during training. We also mask a subset of the tape tokens in the bank randomly. To implement this, we initialize $m = 0 + b$ for ATR algorithm, where $b \in \mathbb{R}^{1 \times C}$ and $b_c \sim \text{Bernoulli}(p)$. We set $p$ as 0.1 by default. We also observed that larger $p$ can further improve the training stability.

### A.8 MORE RESULTS ON IMAGENET

Table 6: Results of training on ImageNet-1K only. We use the input-dirven bank for AdaTape. * denotes that we approximate the throughput by the models with similar architecture and input pipeline. For A-ViT, we not only report their results from the paper but also re-implement A-ViT by training from scratch, *i.e.,* A-ViT(Ours).

| Model | Adaptive | #Param | Throughput | ImageNet Top-1 (%) |
|---|---|---|---|---|
| ViT-Ti/16 | | 5.7 | 387.5 | 58.7 |
| DeiT-Ti/16 | | 5.7 | 381.8* | 71.3 |
| PlainViT-Ti/16 | | 5.7 | 381.8 | 73.0 |
| U2T-Ti/16 | ✓ | 6.1 | 364.3 | 70.2 |
| A-ViT-Ti/16 | ✓ | 5.7 | 226.6* | 71.0 |
| A-ViT-Ti/16(Ours) | ✓ | 5.7 | 226.6 | 73.2 |
| AdaTape-Ti/16 | ✓ | 6.3 | 380.9 | **73.6** |
| ViT-S/16 | | 22.0 | 385.7 | 75.2 |
| DeiT-S/16 | | 22.0 | 365.8* | 78.9 |
| PlainViT-S/16 | | 22.0 | 365.8 | 79.2 |
| U2T-S/16 | ✓ | 23.8 | 163.2 | 74.5 |
| A-ViT-S/16 | ✓ | 22.0 | 166.6* | 78.6 |
| A-ViT-S/16(Ours) | ✓ | 22.0 | 166.6 | 77.0 |
| AdaTape-S/16 | ✓ | 24.3 | 366.5 | **79.5** |
| PlainViT-B/16 | | 87.1 | 159.9 | 79.5 |
| AdaTape-B/16 | ✓ | 94.9 | 130.5 | **80.8** |

We train with ImageNet-1K only and summarized the results in Table 6. We can see AdaTape outperforms all adaptive baselines by a large margin. Even compared to highly-optimized baselines without adaptivity like PlainViT (Beyer et al., 2022) and DeiT (Touvron et al., 2021), AdaTape can still surpass them with a comparable computation budget. For instance, AdaTape-S/16 uses only 0.4× training cost and 0.3× parameters but achieves almost comparable results with PlainViT-B/16. We may also note that Tiny scale models cannot achieve much higher throughput than Small scale models on ImageNet. We suggest the reason is that the efficiency bottlenecks are mainly from data loading and preprocessing instead of the computation budget within neural networks.

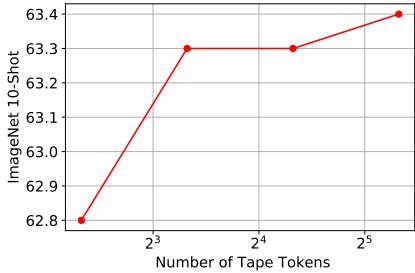

Figure 6: We sweep the number of tape tokens over {5, 10, 20, 40}. Considering more tape tokens mean much more computation cost, we set 10 as our default choice.

### A.9 ABLATION ON HYPER-PARAMETERS

**Number of tape tokens**   We investigate the effect of the number of tape tokens in this section. In the model with adaptive sequence length, the number of tape tokens is controlled by the model adaptively. To control the real length directly, we use the AdaTape without adaptive length as a platform. We sweep the number of tape tokens $T$ over {5, 10, 20, 40} and summarize the results in Figure 6. We can see an obvious improvement when we increase the $T$ from 5 to 10. However, the model is saturated after that. Since using a longer sequence means more computation budget, we select to use 10 tape tokens as the default choice.

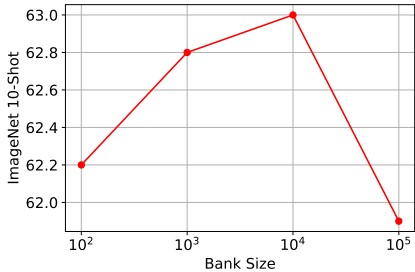

Figure 7: We sweep the bank size over {1e+2, 1e+3, 1e+4, 1e+5} for AdaTape with a learnable bank, and found 1e+4 is the sweep point.

**Bank Size**   We also sweep the bank size over {1e+2, 1e+3, 1e+4, 1e+5} for AdaTape with a learnable bank. As shown in Figure 7, the AdaTape performs best when we have $1e+4$ tape tokens in the bank. As we fix the number of recurrences as 10 in ATR algorithm by default, a larger bank will only increase the computation cost linearly.

### A.10 ABLATION ON COMPUTATION COST

Although AdaTape only increases the computation cost slightly, to further verify the improvement is from more reasonable design and adaptive abilities instead of more computation, we conduct ablation experiments on computation cost. First, as shown in Figure 8, even if a smaller patch size can improve ViT, AdaTape is still outperforming ViT with less computation. In addition, we report the quality-cost comparison in terms of throughput in Figure 9. We can see AdaTape can outperform ViT significantly with smaller latency.

### A.11 ABLATION ON NUMBER OF TRAINABLE PARAMETERS

Since we have two FFNs in every transformer block, AdaTape has more trainable parameters than ViT. To validate that the improvement is not just caused by having more parameters, we increase the

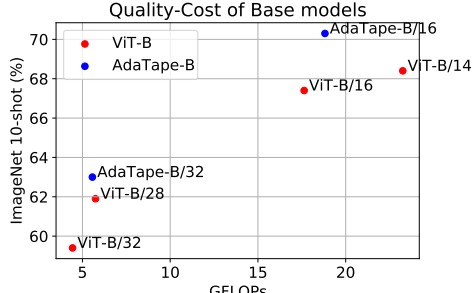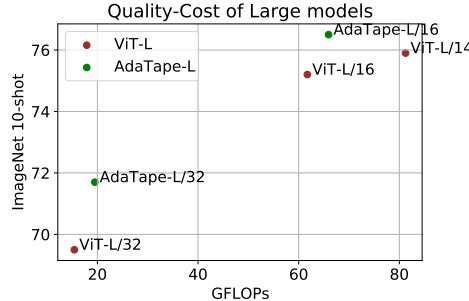

Figure 8: Ablation Study on the quality-cost trade-off. We scale the standard transformer (*i.e.,* ViT) to a smaller patch size, *e.g.,* ViT-B/14 and ViT-L/14, and compare it with AdaTape.

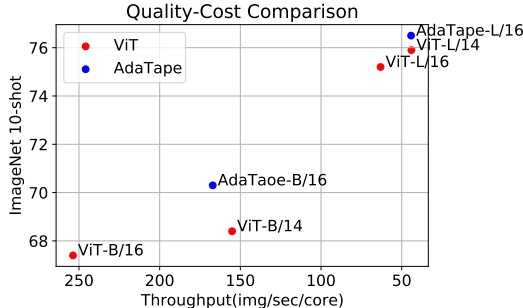

Figure 9: we report the quality-cost comparison in terms of throughput to verify that AdaTape is better than baselines with comparable computation cost.

trainable parameters in ViT by adding 2 more FFN layers to process different tokens. Then, ViT has 3 FFNs in total. The first FFN is used to handle `[CLS]` token. The second FFN and the third FFN are fed by half of the patch tokens, respectively. The results are summarized in Table 7. We can see AdaTape outperforms ViT with 3 FFNs using less computation and fewer parameters. For instance, AdaTape-B/16 surpasses ViT-B/14-3FFN by $1.3\%$ in terms of top-1 accuracy on ImageNet 10-shot.

## A.12 VISUALIZATION OF TOKEN SELECTION DISTRIBUTION

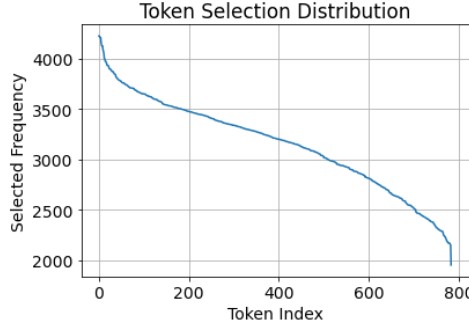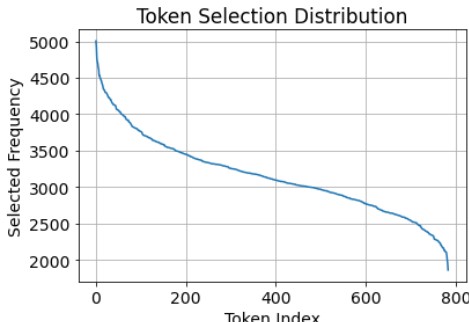

Figure 10: We visualize the tape token selection distribution in AdaTape-B/32 (left) and AdaTape-B/16 (right). The index id is sorted by the value of the selected frequency

Table 7: Ablation study on the number of trainable parameters.

| Model | GFLOPs | Throughput | #Param | IN 10-shot |
|---|---|---|---|---|
| ViT-B/28 | 5.730 | 661.1 | 100.9 | 61.9 |
| ViT-B/28-3FFN | 5.744 | 517.0 | 200.1 | 62.4 |
| AdaTape-B/32 | 5.585 | 431.8 | 185.6 | 63.0 |
| ViT-B/14 | 23.254 | 155.1 | 99.7 | 68.4 |
| ViT-B/14-3FFN | 23.239 | 148.8 | 198.9 | 69.0 |
| AdaTape-B/16 | 18.837 | 167.1 | 192.5 | 70.3 |

Similar to Section 3.5, we collect the token selection results on JFT-300M validation set. We sort the tape token index by the frequency and visualize the token selection distribution in Figure 10. We can observe the token selection decision obey long-tail distribution. That shows our AdaTape prefers the tape tokens at some specific positions, which is similar to our observation in Figure 4, *i.e.,* central patches are frequently selected.

