# OpenReview forum: "Adaptive Computation with Elastic Input Sequence"
_ICLR.cc/2023/Conference — Submitted to ICLR 2023_

### Official Review · Reviewer_vn6P · 2022-10-24

**Confidence:** 4
**Correctness:** 3
**Technical Novelty And Significance:** 4
**Empirical Novelty And Significance:** 3
**Recommendation:** 6

**Clarity, Quality, Novelty And Reproducibility:**

The paper is clear, but contains many typos/grammatical errors. The methods are original, however there is no code or promise to release code. The details in the text might make reproducing the findings difficult, although I did not try to reproduce them myself.

**Strength And Weaknesses:**

This paper proposes a memory bank that can be queried adaptively for a variable number of tokens to append to the input sequence. As a way to add adaptivity to existing neural architectures, this is an alternative to recurrence or weight sharing. There is novelty in this choice, although it isn't adequately contextualized among the existing adaptive methods (listed below). Additionally, the choice of benchmarks is quite odd. While the first task considered, parity, is popular in some early adaptive compute papers, it strikes me as outdated. The latest work -- largely branded as algorithmic reasoning, but with the same focus -- includes methods that can compute prefix sums of binary strings (arguably more work and including the parity task) of many more bits than considered here (listed below). The second task is image classification which is not considered in most other adaptive compute/algorithmic reasoning papers.

If the goal of this work is to study, explore, and improve adaptivity in neural networks, then the choice of benchmarks and competing models is lacking. If the goal is to augment transformer architectures to improve their performance it aught to be better framed in the text.

Strengths:
- Creative and novel method for added adaptivity to transformer models

Weaknesses:
- Missing related work
    - Adpative methods that use recurrence and solve algorithmic tasks like parity:
        - [1] Schwarzschild, Avi, et al. "Can you learn an algorithm? generalizing from easy to hard problems with recurrent networks." Advances in Neural Information Processing Systems 34 (2021): 6695-6706.
        - [2] Veličković, Petar, et al. "The CLRS Algorithmic Reasoning Benchmark." arXiv preprint arXiv:2205.15659 (2022).
        - [3] Ibarz, Borja, et al. "A Generalist Neural Algorithmic Learner." arXiv preprint arXiv:2209.11142 (2022).
    - Adaptive model that use recurrence can compute prefix sums (and therefore parity) of inputs with 512 bits.
        - [4] Bansal, Arpit et al. End-to-end Algorithm Synthesis with Recurrent Networks: Logical Extrapolation Without Overthinking. Neural Information Processing Systems (NeurIPS), 2022.
- Some minor writing issues (not affecting score, only a few examples here):
    - Middle of Page 4: The following excerpt is not full sentences and has incorrect punctuation. "For instance, using a smaller patch size for generating the bank can be seen as dynamic
multi-scale processing of the input where we can select what fine-grained information from the input
is useful which is much more efficient than using all the small patches. , which would consume a
large amount of computation."
    - Bottom of Page 4: "The of-the-shelve ACT ..." should probably be "...off-the-shelf...".
    - Bottom of Page 4: "For instance, as shown in algorithm 1..." where "algorithm" should be capitalized."
    - Bottom of Page 5: "In order to incentives shorter sequences..." should probably be "...incentivise...".
    - The Appendix has bad references to Figures that are not there.


___

In their reply, the authors addressed all my concerns. I have changed my score accordingly.



**Summary Of The Paper:**

This paper describes a novel way to add adaptivity to transformer models. By storing or generating a bank of tokens for each input, the input can be adaptively extended with a variable number of tokens. The method is compared to transformers without adaptivitiy as well as the Universal Transformer. The benchmarks for comparison include computing parity of bit stings as well as image classification.

**Summary Of The Review:**

I find this paper well below the threshold for acceptance. The experiments seem out of tune with the related work. Although the exact results support the very narrow claim that this model is better than one existing model, the contribution to the field is limited by the scope of comparison.

---

> ### Author Response · Authors · 2022-11-18
> **Response to Reviewer vn6P**
>
> We would like to thank the reviewer vn6P for the comments, feedback, and suggestions. Below, we respond to each comment and pointers to the revised parts of the paper.
>
> > Q1: The main goal, contributions, and choice of experiments and tasks
>
> Thank you for the review and the valuable feedback.
> The core goal of this paper is to introduce adaptivity into Transformers that are becoming the most widely used architecture in various domains and applications, including sequence modeling and computer vision.  We explicitly go for such architectures since it has been used for large-scale training and any idea that can bring a better quality/cost trade-off on that scale will be extremely impactful.  We focus on large-scale image recognition tasks and run experiments in transfer learning setup by pre-training on JFT-300M and evaluating on few-shot setup. We also did the training from scratch on ImageNet (added during rebuttal). We added baselines for adaptive computation that is proposed for a similar setup in our comparisons.
> The experiment on the Parity task is mainly a bonus experiment to support the claim that adaptive computation can be also effective when a particular inductive bias is required by the task, while such a task is unsolvable by a vanilla transformer, just showing the superiority of AdaTape from a different perspective.  In general, we acknowledge the interestingness of your suggestion on experimenting with different algorithmic tasks and we believe more exploration of the idea of “adaptivity in input length instead of model depth” in controlled setups and synthetic tasks can bring more insights.
> We appreciate your feedback and suggestion. In the revised version of the paper, we made our goal more clear and added more experiments and baselines that are more in line with our main target.  We explained that our experimental setup is mainly designed to show the value of adaptivity in a large-scale application setup with a scalable architecture, but also provide some insight on how the new inductive biases can help a Transformer model capable of solving a difficult task that was failing to do before.
>
> > Q2: Related works
>
> Thanks for the suggestion. We added the missing related work to Section 3.2. Besides, we add a new section dedicated to the related work in the revised paper that covers works with similar goals to AdaTape.
>
> > Q3: Writing
>
> Thanks for the detailed comments on the writing.  We proofread the paper, improved the writing, and fixed the typos.

---

> > ### Comment · Reviewer_vn6P · 2022-11-21
> > **Follow up to Authors' response**
> >
> > Thanks for the thorough response and the updates to the draft. In the revised draft my concerns have all been addressed. Upon reading the authors' response and rethinking, I see the value in the large scale image classification experiments that I admit eluded me earlier. I am updated my score to reflect that I think this paper should be accepted.
> >
> > Thank you again to the authors for the effort and clarity in their response.

---

> > > ### Author Response · Authors · 2022-11-21
> > > **Reply to Reviewer vn6P**
> > >
> > > We really appreciate the reviewer for reading our response and the revised paper and also for changing the score. We are happy that the revised draft addresses all the concerns and again acknowledge that the comments from the reviewer have led to improvements in the paper for all the readers. Thanks for your feedback.

---

### Official Review · Reviewer_pZjZ · 2022-10-24

**Confidence:** 5
**Correctness:** 4
**Technical Novelty And Significance:** 4
**Empirical Novelty And Significance:** 3
**Recommendation:** 6

**Clarity, Quality, Novelty And Reproducibility:**

Quality - great
Clarity - mostly clear, however, understanding Algorithm 1 is not easy, having a diagram will help
Novelty - great

**Strength And Weaknesses:**

Strength:

- The motivation of the paper is great. Compute should be adaptive with respect to the complexity of input sequence.
- The method, AdaTape seems to be novel and differences to ACT are clearly mentioned and sufficient.
- Analysis of the effect of LayerNorm on ACT is insightful.
- Large scale image classification task results are great to test methodology.

Weaknesses:
- Comparison to other adaptive networks/methods is missing.
- Main results are presented with the model trained on JFT-300M that is not a public dataset (to my knowledge) and will make comparison in the future hard. Running main experiments on ImageNet-1K (not few shot) will make the work more valuable for future comparisons.
- Not clear if the code will be available for reproducibility.
- More clarification will be helpful for the points below.


Questions:
- How to make sure that elements in $q$ are sorted by importance? Selecting the first $h$ implies that elements are ranked. Algorithm .
- Can the compute be controlled by user? For example, during the inference compute is adapted to the current hardware load. Probably changing the stoping criteria will work.
- Why not to do selection of input tokens instead?
- In 2.1 authors mention that ACT uses a linear component $q$ to compute halting score. I am not sure it is linear as it usually has sigmoid at the output to limit the score in 0..1.
- There works that enable dynamic compute for neural networks. For example, [1], ConvNet-AIG, Dynamic-ViT[3], Avit[2], Adavit[5]. Even more papers are mentioned in [5]. Comparing to these works will help to understand the contribution better.

References:

[1] Michael Figurnov, Maxwell D Collins, Yukun Zhu, Li Zhang, Jonathan Huang, Dmitry Vetrov, and Ruslan Salakhutdinov. Spatially adaptive computation time for residual networks. In CVPR, 2017.

[2] Yin, Hongxu, et al. "A-ViT: Adaptive Tokens for Efficient Vision Transformer." Proceedings of the IEEE/CVF Conference on Computer Vision and Pattern Recognition. 2022.

[3] Rao, Yongming, et al. "Dynamicvit: Efficient vision transformers with dynamic token sparsification." Advances in neural information processing systems 34 (2021): 13937-13949.

[4] Veit, Andreas, and Serge Belongie. "Convolutional networks with adaptive inference graphs." Proceedings of the European Conference on Computer Vision (ECCV). 2018.

[5] Meng, Lingchen, et al. "AdaViT: Adaptive Vision Transformers for Efficient Image Recognition." Proceedings of the IEEE/CVF Conference on Computer Vision and Pattern Recognition. 2022.

[6] Yizeng Han, Gao Huang, Shiji Song, Le Yang, Honghui Wang, and Yulin Wang. Dynamic neural networks: A survey. TPAMI, 2021.


**Summary Of The Paper:**

Paper proposes a new method called AdaTape that enables adaptive computation method. It is applied to vision transformers to achieve dynamic sequence length that will make compute adaptive to the input image. First, a set of learnable or predefined tokens are stored in the Tape bank, then Adaptive tape reader queries tape tokens from it. The method is inspired and built upon ACT proposed for RNNs. The AdaTape is tested on image classification task.

**Summary Of The Review:**

Paper attacks a n important problem of enabling dynamic compute in neural networks. The method is novel and technically sounds. However, have main results on not public dataset limits its contributions. The absence of comparisons to other dynamic compute methods for CNNs/Transformers makes it hard to understand the impact. The code is not available and reproduction of results is difficult. Hopefully, these limitations will be addressed during the rebuttal, I am happy to reconsider my score.

---

> ### Author Response · Authors · 2022-11-18
> **Response to Reviewer pZjZ:**
>
> We would like to thank reviewer pZjZ for the comments, feedback, and suggestions. Below, we respond to each comment and pointers to the revised parts of the paper.
>
> > Q1: Compared to other adaptive networks and Running main experiments on ImageNet-1K
>
> Thanks for the suggestion. Indeed including experiments on a standard dataset makes comparisons easier. We conducted comprehensive experiments on ImageNet-1K by training from scratch and reported the results in Figure 3 and Table 6. We can see AdaTape outperforms one of the most recent baselines with adaptive computation,  A-ViT, by a large margin in terms of both training efficiency and effectiveness. This is achieved by decoupling the adaptivity from model architecture. Since AdaTape injects adaptivity into the input sequence directly, AdaTape is relatively more hardware friendly than other adaptive models like A-ViT and UT. We will release checkpoints of the trained model on ImageNet-1K along with the code.
>
> > Q2: Not clear if the code will be available for reproducibility.
>
> We will add pointers to the open-sourced code and will release checkpoints of trained AdaTape with different sizes on ImageNet1k. Given the difficulty of implementation of adaptive computation ideas (especially baselines we reproduced), we hope our code will contribute to the progress of research in this direction.
>
> > Q3: How to make sure that elements in q are sorted by importance?
>
> Actually, the elements in $q$ are not required to be sorted by importance. We compute the similarity between the slice of $q$ and tape tokens, and we then conduct the top-K selection to take the most similar tokens within the bank. Therefore, we just need to sort the similarity scores for the top-K selection. We made this clear in the revisied paper.
>
> > Q4: Can the compute be controlled by user?
>
> Yes, the computation budget can be controlled by changing the value of $\tau$ in Algorithm 2. When we set a smaller $\tau$, the adaptive reading will halt earlier. Another knob is to set the _max_length_ to a larger or smaller number.
>
> > Q5: Why not do the selection of input tokens instead?
>
> We propose two versions of banks for AdaTape, “learnable bank” and “ input-driven bank”. For AdaTape with an input-driven bank, we select tape tokens from the input which is exactly what you suggested. One main point about AdaTape is that it dynamically “scales” the computational budget instead of dynamically “reducing” the computational cost by pruning input (which is done in UT (U2T) and A-ViT baselines). These two setups (i.e. pruning inputs and AdaTape) are in fact two sides of the same coin, one starting from a model with a less computational budget and adding to it and one starting from a model with a more computational budget and reducing it. We found AdaTape has a much more stable training compared to UT/A-ViT which requires making discrete decisions in each layer and performing better.
>
>
> > Q6: ACT uses a linear component? It also has a sigmoid function.
>
> Thanks for the question. That is correct. We have added this detail to the paper.

---

### Official Review · Reviewer_V6XW · 2022-10-25

**Confidence:** 3
**Correctness:** 3
**Technical Novelty And Significance:** 2
**Empirical Novelty And Significance:** 2
**Recommendation:** 5

**Clarity, Quality, Novelty And Reproducibility:**

There are quite a few grammatical errors starting from Section 2.2 that make the paper difficult to understand.


**Strength And Weaknesses:**

Strength:
1. The proposed AdaTape not only adaptively selects different parameters to handle different inputs, but also adjusts the computational budget by using variable-length storage tapes.
2. The introductory part is well written and engaging.

Weaknesses:
1. There are quite a few grammatical errors starting from Section 2.2 that make the paper difficult to understand.


**Summary Of The Paper:**

In this paper, the authors investigate the problem of adaptive computation time and propose AdaTape, which can adaptively select different function types and computation budgets for different inputs. It incorporates an adaptive tape reader (ATR) to learn or extract variable-length tape tokens for dynamic computation in neural networks. The authors conducted experiments on the parity task and the image classification task. Experimental results show that the proposed AdaTape algorithm outperforms several baselines such as ViT, Universal Transformers, and their variants, while maintaining the same training efficiency.

**Summary Of The Review:**

The writing (except the introduction section) needs to be significantly improved to enhance readability. It is recommended to add more clarification on the purpose of the tape token and compare with other possible methods that can achieve the same goal. There is no separate related work section. It may be helpful to add that section.

---

> ### Author Response · Authors · 2022-11-18
> **Response to Reviewer V6XW**
>
> We would like to thank reviewer V6XW for the comments, feedback, and suggestions. Below, we provide responses to each comment and pointers to the revised parts of the paper.
>
> > Q1: Grammar errors
>
> Thanks for the careful review and for pointing it out. We have proofread the draft, improved the writing, and fixed grammatical errors carefully in our new revision.
>
> > Q2: The purpose of tape token
>
> A very high-level motivation for adding tape tokens is supporting additional dynamic “read and write” tape for the algorithm, which would provide more computational capacity on demand.
> In a more technical view, adding tape tokens means adding to the FLOPs per example without changing the parameters involved in the computation of the model.
> In AdaTape, tape token is combined with adaptivity and due to the sparse tape token selection in our design, the larger capacity and adaptive input content provided by tape tokens would make AdaTape outperform the baselines (Table 2). We added a bit of this discussion to the third paragraph of the introduction.
>
> > Q3: Other methods being able to achieve the same goal
>
> To the best of our knowledge, AdaTape is the first work introducing adaptivity on the input sequence, as opposed to all previous works that use adaptivity through variable model depth or sparse/conditional computation. Section 2.1 and 2.3. discuss specifically how well-established algorithms and ideas, like ACT, can be used for adaptive sequence length. We found there are some key contradictions between their assumptions and what is needed for adaptive sequence length. We then propose ATR in Section 2.4 which is completely new and effective in our setup. Note that adaptive sequence length can be used as an orthogonal idea to the existing adaptive computation ideas and we plan to explore this direction more in the future.
>
> > Q4: Related work
>
> We added a related work section in the revision that is dedicated to covers and groups related works and their connections to AdaTape, which can be found in Appendix.

---

### Official Review · Reviewer_rSKm · 2022-10-31

**Confidence:** 3
**Correctness:** 4
**Technical Novelty And Significance:** 3
**Empirical Novelty And Significance:** 2
**Recommendation:** 5

**Clarity, Quality, Novelty And Reproducibility:**

The clarity of the paper can be improved. In particular, the experimental evaluation should contain more details regarding the training and evaluation on the datasets other than JFT-300M.

**Strength And Weaknesses:**

Strengths
-----------

- Adaptive computation is a very interesting and important subject, both in terms of saving computation by only computing as much as it is necessary as well as mimicking the human thought process of considering a problem again and again and collecting more information.
- Adding extra tokens from a token bank is a very intuitive way to inject variable sized pieces of information to a transformer model.

Weaknesses
---------------

- AdaTape is significantly slower than the baselines often being 2 times slower. This significantly reduces the usefulness of the method since the baselines with a few extra layers would probably perform as well.
- The experimental evaluation is not very clearly written. In particular the information about finetuning the datasets except for JFT-300 is missing.
- The usefulness of the adaptive computation abilities of AdaTape is put in question from the experiments in section 3.4 . In particular we see that in most cases, not using the adaptive length algorithm actually yields better results. It would also be significantly faster on GPUs and TPUs.

**Summary Of The Paper:**

The paper proposes a novel method to add "on demand" computation to transformers by selecting additional tokens from a token bank. In particular the authors use a variation of the adaptive computation time algorithm used in universal transformers to add additional tokens in the input sequence. Experiments with image classification shows that the proposed method can indeed make use of the extra computation to improve the classification results.

**Summary Of The Review:**

The paper seems a great relatively novel idea, however the experimental evaluation is lacking. The comparisons show that using the proposed method is actually significantly slower albeit better in terms of final accuracy. However, not using the adaptive tape reading algorithm actually performs better and would be possibly faster in modern accelerators.

---

> ### Author Response · Authors · 2022-11-18
> **Response to Reviewer rSKm**
>
> We would like to thank the reviewer rSKm for the comments, feedback, and suggestions. Below, we provide responses to each comment and pointers to the revised parts in the paper.
>
> > Q1: AdaTape is slower than ViT.
>
> AdaTape is indeed slightly slower than a ViT with a similar size since AdaTape may add extra tokens to the input that increase total computation per example.  However, we would like to highlight that a fair comparison here would be in terms of the cost-performance trade-off. To make this comparison more clear, we added an ablation study (Fig. 9) in Appendix A.10, with models of different costs and different quality which shows how AdaTape does the trade-off vs baseline. We can see AdaTape outperforms ViT significantly even if we scale ViT to be more computationally expensive and slower than AdaTape.
>
> > Q2: The information about fine-tuning the datasets except for JFT-300 is missing.
>
> We strictly followed the setup of the original [ViT](https://arxiv.org/abs/2010.11929) and [big transfer](https://arxiv.org/abs/1912.11370) paper for a fair comparison. We added detailed information on the setup in Section 3.1 in the revised paper (highlighted that with red color in the latest revision). We also added another set of experiments, i.e. training on ImageNet from scratch, for easier comparison in future work.
>
> > Q3: AdaTape without adaptive length performs better.
>
> This is a great question. We elaborate on this here and added more details to the paper for clarity as well as to address this concern for the final readers.
>
> First of all, we would like to highlight that in terms of accuracy, without adaptive length, AdaTape keeps a comparable performance. For instance, in Table 2, AdaTape-L/16 even outperforms the “AdaTape-L/16 without adaptive length” in all datasets. For the smallest model AdaTape-B/32, after removing the adaptivity in length, the ImageNet 10-shot accuracy also drops by 0.5 points.
> But a more important point here (and where AdaTape without adaptive length performs better) is again thinking about the cost-quality trade-off. The cost of  “AdaTape without adaptive length” is the upper bound of AdaTape, since it uses the maximum length (we highlighted this in our revised draft), while AdaTape depending on the complexity of the example might append fewer tokens and in theory, becomes cheaper.  Table 2 provides the average and variance of sequence length used by AdaTape as some statistics that help how adaptivity can save the cost. Although we show that the AdaTape can save compute by choosing to process a shorter sequence compared to “AdaTape without adaptive length”, this fact is not reflected in the numbers we report for the cost of the model (e.g., throughput and FLOPs). This is because we insist on reporting the real cost in a practical setup [1] (unlike many previous works that report the potential/theoretical cost with batch size==1).  As a matter of fact, the main idea of AdaTape, (adaptivity in input length instead of model depth) is much more effective and simple compared to the adaptive computation baselines and we can see a significantly better quality-cost trade-off from AdaTape in Figure 3 (which is added in the revised version of the paper for more clarity on this discussion).
>
> ——————
>
> [1] The reason is that implementing variable sequence lengths with batch-size bigger than one was not possible on the current hardware that we experimented with (TPUs), thus we pad sequences up to max length and mask out the computation that could be saved if we had hardware supporting this setup. This is an issue with all adaptive computation models that use less computation per example as they never had the winning “[hardware lottery](https://arxiv.org/abs/2009.06489)” ticket.

---

### Author Response · Authors · 2022-11-18
**General Response**

Dear reviewers and AC:

We thank all the reviewers for their feedback. We have revised the paper to address questions and comments. The change log is summarized below.

* We proofread the paper carefully, made the writing different parts of the paper more clear, and fixed the typos.

* We added more experiments about training on ImageNet from scratch and present the results of AdaTape vs baseline methods (Figure 3) on this standard benchmark.  We will release the checkpoints from these experiments to the public for further experimentation.

* We added the state-of-the-art adaptive computation baselines from literature like [A-ViT](https://a-vit.github.io/) to make a more comprehensive comparison. Results show that AdaTape can outperform the adaptive baselines on both effectiveness (in terms of accuracy) and efficiency (in terms of throughput). Results are presented in Section 3.3 and Figure 3.

* We added another quality-cost trade-off curve in Figure 9 to show AdaTape can be both faster and more accurate than baselines.

* We added more explanations to further clarify different parts, including the motivations and detailed setups in different experiments. The relevant modifications are summarized in the response to the corresponding reviewers.

We respond  to comments and questions from each reviewer in more detail and provide pointers to the related parts of the revised paper.

Best,

Authors

---

### Author Response · Authors · 2022-11-29
**Thanks to All Reviewers and Looking Forward to Further Discussion**

Dear reviewers,

We would like to thank all the reviewers for taking the time to give feedback on our work.
We put our best effort to address the comments and answer the questions.  We appreciate that some reviewers replied already. For others, we are happy to discuss further for any clarification needed or any point that helps reviewers to consider revising their score.



Best regards,

The Authors

---

### Decision · Program_Chairs · 2023-01-20

**Decision:**

Reject

**Justification For Why Not Higher Score:**

Reviewers had concerns on the lack of sufficient comparison with existing different adaptive methods, and the writing that would need more polishing and clarity on experimentation and empirical conclusions.

**Justification For Why Not Lower Score:**

 The work is well motivated. The method is sound. Results show improved accuracy with the same training efficiency compared to certain baselines.

**Metareview: Summary, Strengths And Weaknesses:**

The paper proposes a new method AdaTape that enables adaptive computation in Transformers. Specifically, the method constructs a token bank with a set of learnable/predefined tokens, and selects tokens from the token bank with an adaptive tape reader to achieve dynamic sequence length according to the input sequences. The method is applied to vision Transformer for image classification and shows improved performance without added computation time. The work is well motivated. The method is sound. Reviewers had concerns on the lack of sufficient comparison with existing different adaptive methods, and the writing that would need more polishing and clarity on experimentation and empirical conclusions.